# Ex Vivo Infection of Human Placental Explants by *Trypanosoma cruzi* Reveals a microRNA Profile Similar to That Seen in Trophoblast Differentiation

**DOI:** 10.3390/pathogens11030361

**Published:** 2022-03-16

**Authors:** Lisvaneth Medina, Jesús Alejandro Guerrero-Muñoz, Ana Isabel Liempi, Christian Castillo, Yessica Ortega, Alfredo Sepúlveda, Fernando Salomó, Juan Diego Maya, Ulrike Kemmerling

**Affiliations:** 1Programa de Anatomía y Biología del Desarrollo, Instituto de Ciencias Biomédicas, Facultad de Medicina, Universidad de Chile, Santiago 8380453, Chile; lisvanethmedina@hotmail.com (L.M.); jesus.guerrero@ug.uchile.cl (J.A.G.-M.); anitavet@gmail.com (A.I.L.); ccastillor@uchile.cl (C.C.); alfredo.sepulveda.c@ug.uchile.cl (A.S.); fsalomo@gmail.com (F.S.); 2Facultad de Medicina Veterinaria y Agronomía, Universidad de Las Américas, Santiago 7500975, Chile; 3Laboratorio de Enzimología de Parásitos Facultad de Ciencias, Universidad de los Andes, Mérida 5105, Venezuela; andreina898@hotmail.com; 4Programa de Farmacología Molecular y Clínica, Instituto de Ciencias Biomédicas, Facultad de Medicina, Universidad de Chile, Santiago 8380453, Chile; jdmaya@uchile.cl

**Keywords:** *Trypanosoma cruzi*, miRNAs, placenta, trophoblast differentiation

## Abstract

Congenital Chagas disease, caused by the protozoan parasite *Trypanosoma cruzi*, is responsible for 22.5% of new cases each year. However, placental transmission occurs in only 5% of infected mothers and it has been proposed that the epithelial turnover of the trophoblast can be considered a local placental defense against the parasite. Thus, *Trypanosoma cruzi* induces cellular proliferation, differentiation, and apoptotic cell death in the trophoblast, which are regulated, among other mechanisms, by small non-coding RNAs such as microRNAs. On the other hand, ex vivo infection of human placental explants induces a specific microRNA profile that includes microRNAs related to trophoblast differentiation such as miR-512-3p miR-515-5p, codified at the chromosome 19 microRNA cluster. Here we determined the expression validated target genes of miR-512-3p and miR-515-5p, specifically human glial cells missing 1 transcription factor and cellular FLICE-like inhibitory protein, as well as the expression of the main trophoblast differentiation marker human chorionic gonadotrophin during ex vivo infection of human placental explants, and examined how the inhibition or overexpression of both microRNAs affects parasite infection. We conclude that *Trypanosoma cruzi-*induced trophoblast epithelial turnover, particularly trophoblast differentiation, is at least partially mediated by placenta-specific miR-512-3p and miR-515-5p and that both miRNAs mediate placental susceptibility to ex vivo infection of human placental explants. Knowledge about the role of parasite-modulated microRNAs in the placenta might enable their use as biomarkers, as prognostic and therapeutic tools for congenital Chagas disease in the future.

## 1. Introduction

Chagas disease, also known as American trypanosomiasis, is caused by the protozoan parasite *Trypanosoma cruzi* (*T. cruzi*), and is one of the 13 most neglected tropical diseases [1,2,3]. The parasite can be transmitted from mother to child, causing congenital CD with potential long-term consequences. The World Health Organization estimates that 1,125,000 women of fertile age are infected with *T. cruzi*, with an incidence of congenital infection of 8668 cases/year in the 21 Latin American countries where Chagas Disease is currently endemic [4]. Furthermore, congenital transmission is responsible for 22.5% of new annual infections in countries with controlled domestic vector infestations [4,5]. In addition, migrations of Latin American people have promoted Chagas disease as a global disease, which is now observed in non-endemic areas. Notably, the congenital route is the primary mode of transmission of *T. cruzi* in non-endemic regions, over blood transfusions and organ transplantations [6,7].

Congenital Chagas disease results from complex interactions between the parasite (virulence), the placental barrier’s strengths and weaknesses, and maternal and fetal/neonatal immune responses [2]. The placenta presents local defense mechanisms, including the epithelial turnover of the trophoblast, which is the first fetal tissue that comes into contact with parasites circulating in maternal blood [2,8,9]. The trophoblast is a bistratified epithelium composed of the superficial syncytiotrophoblast (ST) and the basal cytotrophoblast (CT). The cytotrophoblast displays highly proliferative properties, whereas the differentiated syncytiotrophoblast loses its proliferating capacity. The ST is multinucleated, continuous, and usually uninterrupted, covering all villous trees of the human placenta [8,10]. Trophoblast turnover involves the precise orchestration of different cellular processes that include cell proliferation of the cytotrophoblast, cell differentiation (syncytial fusion by incorporating cytotrophoblast cells into the syncytiotrophoblast, and the differentiation of cytotrophoblast cells’ previous fusion with the syncytiotrophoblast), and programmed cell death [10,11].

All these cellular processes imply profound gene expression changes, regulated at the transcriptional or post-transcriptional levels. Non-coding RNAs (ncRNAs), particularly microRNAs (miRNAs), play essential roles in regulating gene expression at the post-transcriptional level [12,13]. MiRNAs repress mRNAs in a sequence-specific manner through mRNA degradation or inhibition of the translation of mRNAs [13,14]. Moreover, parasites, including *T. cruzi*, modulate the host´s gene expression to avoid the various defense mechanisms and establish infection. The alteration of the host’s miRNA expression is one strategy to accomplish this [15].

Interestingly, the largest miRNA cluster in humans is encoded in the chromosome 19 cluster (C19MC) (19q13.41), which is almost exclusively expressed in the placenta [16,17] and particularly in the trophoblast [16]. Furthermore, C19MC-derived miRNAs have been associated with placental development, pregnancy-related pathologies, and infections [13,18,19].

MiR-512-3p promotes trophoblast differentiation by repressing the caspase 8 inhibitor c-FLIP (cellular FLICE-like inhibitory protein); and, consequently, increases caspase 8 activity [20] (Figure 1). Caspase 8 regulates trophoblast differentiation and apoptotic cell death and is activated by *T. cruzi* [21]. On the other hand, miR-515-5p inhibits trophoblast differentiation by directly repressing the human glial cells missing 1 transcription factor (hGCM-1) gene [22] (Figure 1). Particularly, the hGCM-1 transcription factor mediates syncytin and human chorionic gonadotrophin (hCG) expression [11], both of which are induced by *T. cruzi* in human placental explants (HPE) [8,9]. In addition, *T. cruzi* decreases miR-515-5p and increases miR-512-3p, suggesting that they play an essential role in local placental defense mechanisms against the parasite [21].

In this study, we aimed to determine the expression of miR-512-3p- and miR-515-5p-regulated genes, specifically hGCM-1 and c-FLIP, as well as the expression of the main trophoblast differentiation marker hCG during ex vivo infection of HPE, and examined how inhibition or overexpression of both miRNAs affects parasite infection.

## 2. Results

### 2.1. HPE Can Be Effectively Transfected with miR-512-3p and miR-515-5p Mimics and Antagomirs without Causing Tissue Damage

HPE was transfected with 100 nM of miR-512-3p or miR-515-5p antagomirs (A-512 or A-515), mimics (M-512 or M-515), and their respective scrambles (AS or MS) as described above (Figure 2). Transfection with A-512 (Figure 2A) and A-515 (Figure 2B) decreased the corresponding miRNAs levels in the tissue samples by 61.57% ± 26.00% (*p* ≤ 0.01) and 92.59% ± 1.09% (*p* ≤ 0.01), respectively. On the other hand, M-512 (Figure 2A) and M-515 (Figure 2B) increased the miR-512-3p and miR-515-5p levels in HPE by 69.33% ± 19.27% (*p* ≤ 0.001) and 76.96% ± 20.23% (*p* ≤ 0.001), respectively. As expected, transfection with the respective scrambles did not significantly affect the miR-512-3p or miR-515-5p levels. We further transfected the HPE with a Cy3-conjugated off-target antagomir (Antagomir-Cy3) to detect the tissue localization of the transfect. As shown in Figure 3, the transfect can be observed in the whole placental explant but is mainly located in the trophoblast (white arrows).

In addition, we analyzed the transfected HPE for tissue damage (Appendix A). Transfection did not alter the structure of the chorionic villi.

### 2.2. T. cruzi Increases and Decreases miR-512-3p and miR-515-5p Levels in HPE, Respectively

HPE was transfected with A-512 or AS, M-515, or MS for 24 h and then challenged with 10^5^ parasites/mL for 2 h. The parasite alone (56.65% ± 7.97%; *p* ≤ 0.01) or in the presence of the antagomir scramble (AS) (68.64% ± 19.19%; *p* ≤ 0.01) increased miR-512-3p levels significantly compared to the non-transfected control samples. In the case of A-512 transfected samples, miR-512-3p levels were significantly decreased in the presence (96.15% ± 0.78%; *p* ≤ 0.001) and absence (92.59% ± 1.09%; *p* ≤ 0.001) of the parasite (Figure 4A).

On the other hand, *T. cruzi* decreased the miR-515-5p levels (43.41% ± 26.00%; *p* ≤ 0.01) significantly compared to non-transfected control HPE. However, despite the fact that transfection alone increases miR-515-5p levels (58.03 ± 10.09%; *p* ≤ 0.001), M-515 cannot restore the miR-515-5p levels to control levels (Figure 4B).

### 2.3. T. cruzi Does Not Affect c-FLIP Expression

The caspase 8 inhibitor c-FLIP is a validated target for miR-512-3p [20]. Thus, HPE was transfected and challenged with *T. cruzi* cell-culture-derived trypomastigotes. Here we confirmed that the inhibition and overexpression of miR-512-3p, respectively, increased (38.55% ± 28.88%; *p* ≤ 0.05) and decreased (31.14% ± 14.64%; *p* ≤ 0.05) c-FLIP expression levels (Figure 5A). However, the parasite did not alter c-FLIP expression levels (Figure 5B).

### 2.4. miR-515-5p and miR-512-3p Regulate T. cruzi-Induced Increase in hGCM-1

The transcription factor hGCM-1 gene is a target of miR-515-5p and mediates the transcription of genes involved in trophoblast differentiation [22]. Thus, HPE was transfected and challenged with *T. cruzi* trypomastigotes or Forskolin (100 uM) as a positive control for trophoblast differentiation. Transfection with A-515 increased (54.13% ± 27.57%; *p* ≤ 0.05) and M-515 decreased (52.73% ± 21.72%; *p* ≤ 0.001) hGCM-1 expression (Figure 6A). Concordantly with previous results indicating that *T. cruzi* induces trophoblast differentiation [9], the parasite increased hGCM-1 expression (90.79% ± 44.10%; *p* ≤ 0.0001), an effect that was prevented by miR-515-5p overexpression. However, transfection with A-515 (68.47% ± 23.24% *p* ≤ 0.001) or AS (60.17% ± 32.01% *p* ≤ 0.01) did not further augment the parasite-induced increase in hGCM1 (Figure 6B).

Interestingly, miR-512-3p also modulates hGCM-1 expression levels. Thus, transfection with A-512 decreases (37.26% ± 17.03%; *p* ≤ 0.01) hGCM-1 expression (Figure 6C). However, A-512 transfection was not able to prevent the parasite-induced increase of hGCM-1; moreover, it increased hGCM-1 expression in the presence of *T. cruzi* (83.07% ± 27.60%; *p* ≤ 0.0001), similarly to that induced by forskolin (59.01% ± 20.64%; *p* ≤ 0.0001). On the contrary, transfection with M-512 or MS prevented the parasite-induced increase in hGCM-1 (Figure 6D).

### 2.5. miR-512-3p and miR-515-5p Regulate T. cruzi-Induced Increase in hCG

hCG is one of the main trophoblast differentiation markers [23], induced by the parasite [9], and its expression is mediated by hGCM-1 [11]. Therefore, we evaluated if miR-512-3p and miR-515-5p are involved in parasite-induced hCG expression. Thus, M-512 (145.35% ± 21.21%; *p* ≤ 0.0001) (Figure 7A) and A-515 (72.11% ± 7.95%; *p* ≤ 0.001) (Figure 7C) increased hCG expression, respectively. Despite the fact that A-512 and M-515 did not alter the respective miR-512-3p and miR-515-5p levels, A-512 prevented the *T. cruzi*-induced increase in hCG expression (233.55% ± 28.98% *p* ≤ 0.0001) but M-512 did not further increase the parasite-induced increase in hGC (230.82% ± 46.84% *p* ≤ 0.001) (Figure 7B). On the other hand, both M-515 (212.89% ± 16.13% *p* ≤ 0.05) and A-515 (238.52% ± 58.62% *p* ≤ 0.05) partially prevented the parasite-induced increase in hCG expression (379.01% ± 67.47% *p* ≤ 0.0001) (Figure 7D).

### 2.6. miR-512-3p and miR-515-5p Levels Determine HPE Susceptibility to T. cruzi Infection

Trophoblast differentiation, as a part of trophoblast epithelial turnover, is considered a local defense mechanism against *T. cruzi* infection [8,9,21]. We therefore analyzed whether interfering with this cellular process through miR-512-3p and miR-515-5p levels changed parasite infection levels. As expected, by inhibiting miR-512-3p with A-512, and therefore impairing trophoblast differentiation, the parasite DNA load increased significantly (175.60% ± 35.83%; *p* ≤ 0.0001) (Figure 8A). Interestingly, either inhibition (73.10% ± 19.05% (*p* ≤ 0.01)) or overexpression (80.75% ± 40.07%; *p* ≤ 0.01) of miR-515-5p increased *T. cruzi* DNA load in HPE (Figure 8B).

## 3. Discussion

MiRNAs regulate the expression of more than 30% of fundamental genes that, in turn, are involved in critical biological processes, including cellular differentiation and immune responses, determining the success or failure of infection [15,24,25].

During host–pathogen interactions, different pathogens, including protozoan parasites, modulate the host´s gene expression to avoid its clearance and to establish infection [15,26].

In HPE, *T. cruzi* not only induces a specific mRNA profile [27], it also induces a specific miRNA profile that is different from that of another protozoan parasite, *Toxoplasma gondii* (*T. gondii*), which might explain the low congenital transmission rates observed for *T. cruzi* and the high transmission rates for *T. gondii* [15]. In that previous study, miR-512-3p and miR-515-5p were validated as miRNAs modulated by *T. cruzi* and this finding was concordantly confirmed in the present work (Figure 4). Both miRNAs are interesting molecules since they are encoded in chromosome 19 (C19MC) (19q13.41), which is the largest miRNA cluster in humans and is almost exclusively expressed in the placenta, undifferentiated embryonic stem cells, and germ cells [16,17]. Furthermore, C19MC-derived miRNAs have been associated with placental development [13], pregnancy-related pathologies, and infections [13,16,28,29]. This organ-specific expression of the studied miRNAs makes them attractive molecules for further studies as diagnostic, prognostic, or therapeutic tools for infections. Notably, C19MC miRNAs are among the most abundant miRNAs expressed in human trophoblastic cells, at least in term placenta [17].

Both miRNAs regulate trophoblast differentiation [20,22], a cellular process related to trophoblast epithelial turnover, considered a local placental defense mechanism against *T. cruzi* [2,8,9,21,30]. The epithelial turnover is considered part of the innate immune system since pathogens, before cell invasion, must attach to the surface of cells. As these cells are continuously eliminated, the attached pathogens are removed [8,31]. Moreover, infected epithelial cells have an alarm system to alert uninfected neighboring cells by transferring danger signals via the gap junction, enabling the epithelium to dispose of infected host cells [31,32]. Danger signals are propagated via nuclear factor kappa B (NFkB) and mitogen-activating protein kinase (MAPK) signaling, pathways that *T. cruzi* activates in HPE [33,34].

MiR-512-3p regulates trophoblast differentiation by repressing the caspase 8 inhibitor c-FLIP, leading to increased caspase 8 activity [20]. Caspase 8 regulates trophoblast differentiation and apoptotic cell death and is activated by *T. cruzi.* Moreover, caspase 8 inhibition promotes parasite infection in BeWo cells, evidenced by an increase in the parasite DNA load and the number of parasites per cell [21]. Here, we confirmed that, in HPE, c-FLIP expression is regulated by miR-512-3p, but its expression is not altered by *T. cruzi* (Figure 5). In addition, we also confirmed that miR-515-5p regulates its validated target hGCM-1 in HPE and that the parasite modulates its expression (Figure 6A,B). However, transfection with the miR-515-5p mimic could not prevent the increase in parasite-induced hGCM-1 expression, nor did the inhibition with the antagomir augment its expression. Interestingly, the miR-512-3p antagomir was able to decrease hGCM-1 in spite of the fact that hGCM-1 is not its direct target. However, similarly to the results observed for miR-515-5p, the antagomir could not prevent the parasite-induced increase in hGCM-1 and the miR-512-3p mimics did not alter the *T. cruzi*-induced increase in hGCM-1 expression (Figure 6C,D). On the other hand, miR-512-3p mimic (Figure 7A) and miR-515-5p antagomir (Figure 7C) transfection increased hCG expression, but only the miR-512-3p antagomir (Figure 7B) could completely prevent the parasite-induced increase in the main trophoblast differentiation marker. Interestingly, both miR-515-p mimics and antagomirs partially prevented the *T. cruzi*-induced increase in hCG. It is probable that, in order to prevent the parasite-induced increase in hGCM-1 and hCG, other miRNAs that share the same seed sequence or target gene are needed, as has been reported for C19MC cluster miRNAs [13,35]. For instance, eight miRNAs belonging to the C19MC cluster are upregulated (miR-521, -520h, -517c, -519d, -517d, -542-3p, -518e, and -519a) in women with preeclampsia [33,36], and hGCM-1 is targeted by at least two other miRNAs (miR-106a and miR-19b) [37], providing evidence for the complex regulation of mRNA expression by miRNAs. In addition, miR-515-5p also represses aromatase P450 (hCYP19A1) [37] and frizzled 5 (Fzd5) [38] genes. In the placenta, aromatase P450 catalyzes the synthesis of estrogens from C_19_ steroids that promote trophoblast differentiation in an autocrine manner [37], and Fzd5 is part of the hetero-dimeric receptor family that binds secreted Wingless (Wnt) proteins and promote cell cycle progression and differentiation through the canonical Wnt signaling pathway [35].

Regarding the susceptibility to parasite infection, as expected, the inhibition of miR-512-3p and increased miR-515-5p levels augmented the *T. cruzi* DNA load in the HPE (Figure 8). Thus, impairing trophoblast differentiation through both miRNAs (Figure 1), the parasite can more easily overcome the placental barrier [8]. Interestingly, the inhibition of miR-515-5p also increased parasite infection (Figure 8B). MiR-515-5p regulates cellular proliferation through the Wnt pathway [38] as well as Notch1 (neurogenic locus notch homolog protein 1), and its deregulation is related to tumorigenesis [39], the promotion of bacterial growth, and disbalance of the gut microbiota [40]. Importantly, cellular proliferation is also a fundamental part of the trophoblast epithelial turnover process [2,8,10], and its de-regulation alters the normal physiology of the placental barrier. Therefore, any change in the anatomical placental barrier might lead to an increase in parasite infection.

Importantly, knowledge about the role of parasite-modulated microRNAs in the placenta might allow their use as biomarkers in the future, providing prognostic and therapeutic tools for congenital Chagas disease.

## 4. Materials and Methods

### 4.1. Cell Cultures

VERO cells (ATCC^®^ CCL-81, American Type Culture Collection, New York, NY, USA) were grown in RPMI medium supplemented with 5% fetal bovine serum (FBS) and antibiotics (penicillin-streptomycin). BeWo cells (ATCC^®^ CCL-98, American Type Culture Collection, New York, NY, USA) were grown in DMEM-F12K medium supplemented with 10% FBS, L-glutamine, and antibiotics (penicillin-streptomycin). Cells were grown at 37 °C in a humidified atmosphere with 5% CO_2_, replacing the culture medium every 24 h [9].

### 4.2. Parasite Culture and Harvesting

Semiconfluent VERO cells were incubated with a culture of Y strain epimastigotes (a non-infective cellular form of the parasite) in the late stationary phase, containing about 5% of infective trypomastigotes. Trypomastigotes invade fibroblasts and replicate intracellularly as amastigotes. After 72 h, amastigotes transform into trypomastigotes, which lyse the host cells. The parasites were recovered via low-speed centrifugation (500× *g*), producing trypomastigotes in the supernatant and amastigotes in the sediment [41].

### 4.3. Human Placental Explant (HPE) Culture and Parasite Infection

Ten human-term placentas were obtained from uncomplicated pregnancies from vaginal or cesarean deliveries. Informed consent for the experimental use of the placenta was provided by each patient as stipulated by the Code of Ethics of the Faculty of Medicine of the University of Chile and Servicio de Salud Metroplitano Norte (Approval number AE 010/2019). The exclusion criteria for the patients were the following: major fetal abnormalities, placental tumor, intrauterine infection, obstetric pathology, or any other maternal disease. The organs were collected in a cold, sterile saline-buffered solution (PBS) and processed no more than 30 min after delivery. The maternal and fetal surfaces were discarded, and villous tissue was obtained from the central part of the cotyledons. The isolated chorionic villi were washed with PBS to remove blood, dissected into approximately 0.5 cm^3^ fragments, and co-cultured with *T. cruzi* trypomastigotes (1 × 10^5^/mL) for 2 h or forskolin (100 uM; as a positive control for trophoblast differentiation) in 1 mL of RPMI culture medium supplemented with complement-inactivated FBS and antibiotics [9,42].

### 4.4. HPE Transfection with miRNAs Mimics and Antagomirs

HPE samples were transfected with 100 nM of mimics or antagomirs of miR-512-3p or miR-515-5p and their respective negative controls by incubating them in 1 mL of RPMI culture medium (without phenol red, supplemented with complement-inactivated FBS and antibiotics) for 24 h at 37 °C [43]. Following transfection, HPE was co-cultivated with *T. cruzi* trypomastigotes as described above. HPE was collected in RNAlater solution (ThermoFisher Scientific^®^, Burlington, ONT, Canada), stored at 4 °C for 24 h and later at −80 °C for posterior miRNA and mRNA isolation. In addition, explants were held at 4 °C in 75% ethanol for DNA extraction. Tissue localization and effective transfection were determined by detecting fluorescently-labeled molecules. Thus, HPE was transfected as described above with 100 nM of a Cy3-conjugated off-target antagomir (Antagomir-Cy3) (abm^®^ Richmond, BC, Canada) [43], and transfection efficiency was determined by means of fluorescence microscopy [44]. The supernatant in which the different experimental conditions were incubated was kept at −20 °C until processing to evaluate lactate dehydrogenase activity.

### 4.5. Lactate Deshydrogenase Activity

Enzymatic activity was determined using a Lactate Deshydrogenase Cytotoxicity Detection Kit (Cat# MK401, Takara Bio Inc^®^, Shiga, Japan) as per the manufacturer’s instructions. Briefly, the plates were incubated at room temperature for 30 min in the dark. Then the absorbance was determined with a Varioskan Flash Multimode microplate reader (Thermo Scientific^®^, Waltham, MA, USA) at a reference wavelength of 490 nm. Finally, the results were normalized to the values obtained in the control conditions.

### 4.6. Histology

Routine histological methods were used for tissue analysis of the HPE. Samples were embedded in paraffin and stained with hematoxylin and eosin [42,45].

### 4.7. miRNA and mRNA Expression Analysis via RT-qPCR

HPE was disrupted mechanically in 1 mL RNAzol^®^ RT (Sigma-Aldrich^®^, St. Louis, MO, USA) according to the manufacturer’s instructions to isolate separate fractions of mRNA and microRNA. The isolated product of mRNAs and miRNAs was stored at −80 °C until analysis. A Qubit^®^ Quant-iT™ microRNA Assay Kit (Molecular probes^®^ Eugene, Oregon, USA) and a Qubit^®^ RNA HS Assay Kit (Invitrogen™, Eugene, Oregon, USA) were used to determine the isolated miRNA and mRNA concentrations. To synthesize cDNA of mature miRNAs, a MystiCq™ microRNA cDNA Synthesis Mix Kit (Sigma-Aldrich Merck, St. Louis, MO, USA) was used per the manufacturer’s instructions, whereas a SuperScript™ IV First-Strand Synthesis System (Invitrogen™) was used to obtain mRNAs’ cDNA. Quantification of the cDNA of each miRNA (miR-515-5p and miR-512-3p) was performed using 12.5 µL of 2× MystiCq microRNA SYBR Green qPCR Ready Mix, 0.5 µL of 10 µM of MystiCq Universal PCR Primer, 0.5 µL of 10µM of each specific MystiCq microRNA qPCR Assay Primer (Table 1), 10.5 µL of nuclease-free water, and 1 µL of cDNA in a 25 µL qRT-PCR reaction. For the quantification of C-Flip, hGCM-1, and hCG mRNA, the 20 µL qRT-PCR reaction contained: 10µL SensiFAST™ SYBR® Hi-ROX Kit (Bioline^®,^ Heidelberg, Baden Würtemberg, Germany), 1 µL of each 10nM reverse and forward primer (Table 2), 3 µL nuclease-free water, and 5 µL cDNA. All qRT-PCR reactions were performed in three replicates under the following cycling conditions: initial denaturation at 95 °C for 2 min, followed by 40 cycles of 95 °C for 5 s, 60 °C for 30 s, and a dissociation stage was added, ranging from 60 °C to 95 °C. Gene expressions were calculated using the ΔΔCT relative expression method and normalized to the expression levels of snRNA U6 (RNU6-1) or hGAPDH [46,47].

### 4.8. DNA Amplification via Real-Time PCR

Genomic DNA was extracted from the placental tissue with the Wizard Genomic DNA Purification Kit (Promega^®^, Madison, WI, USA) according to the manufacturer’s instructions and quantified using a µDropPlate in a Varioskan Flash Multimode Reader (Thermo Scientific^®^). For the amplification of human and parasite DNA, two specific pairs of primers were used for hGADPH and *T. cruzi* satellite DNA (Table 3). Each reaction mix contained 0.5 µL at 10 nM of each primer (forward and reverse), 1 ng of DNA from samples, 10 µL of SensiFAST™ SYBR^®^ Hi-ROX Kit (Bioline^®^, Heidelberg, Baden Würtemberg, Germany), and H_2_O for a total of 20 µL. Amplification was performed in an ABI Prism 7300 sequence detector (Applied Biosystems^®^, Foster City, CA, USA). The cycling programs were as follows: initial denaturation at 95 °C for 3 min, followed by 40 cycles of 95 °C for 5 s, 60 °C for 30 s, and a dissociation stage was added, ranging from 60 °C to 95 °C. Relative quantification analysis of the results was expressed as RQ values using the comparative Control (ΔΔCt) method [42].

### 4.9. Statistical Analysis

All experiments were performed in triplicate using at least three different placentas. Results are expressed as means ± S.D, and experimental data were normalized to control values. The significance of differences was evaluated using one-way ANOVA, followed by Dunnett’s post-test.

## 5. Conclusions

Our results suggest that *T. cruzi-*induced trophoblast epithelial turnover, particularly trophoblast differentiation, is at least partially mediated by placenta-specific miR-512-3p and miR-515-5p. In addition, both miRNAs mediate placental susceptibility to ex vivo infection of human placental explants.

## Figures and Tables

**Figure 1 pathogens-11-00361-f001:**
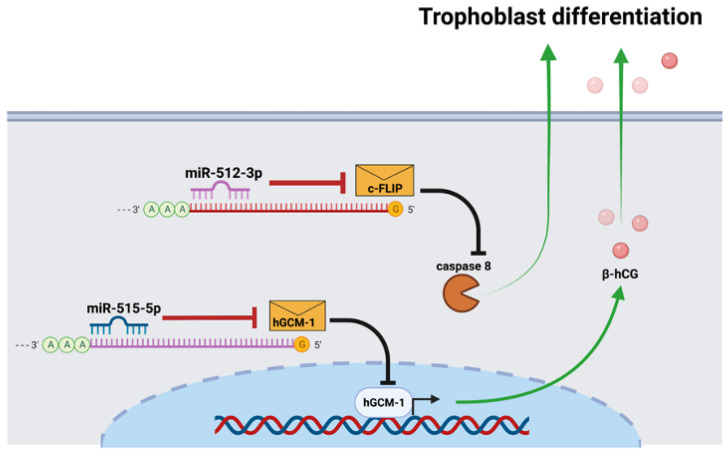
MiR-512-3p- and miR-515-5p = mediated trophoblast differentiation. MiR-512-3p promotes trophoblast differentiation by repressing the caspase 8 inhibitor c-FLIP and, consequently, increases caspase 8 expression, promoting cellular fusion in the trophoblast. Conversely, miR-515-5p inhibits trophoblast differentiation by repressing the transcription factor hGCM-1, which mediates hCG expression.

**Figure 2 pathogens-11-00361-f002:**
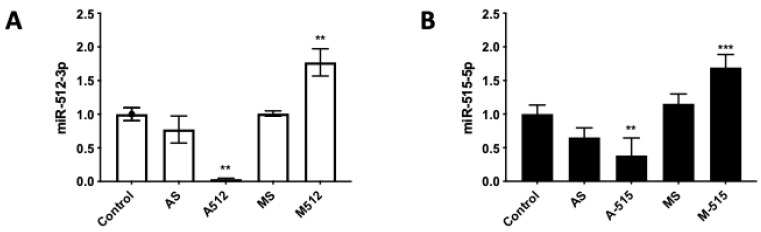
HPE can be effectively transfected with miR-512-3p and miR-515-5p mimics and antagomirs. HPE was transfected with 100 nM of miR-512-3p (**A**) or miR-515-5p (**B**) antagomirs (A-512 or A-515), mimics (M-512 or M-515), and their respective scrambles (AS or MS) for 24 h. The miRNA expression was determined by means of real-time PCR. All values are given as the means ± SD; data were normalized to control values and analyzed via one-way ANOVA and Dunnett’s post-test. ** *p* ≤ 0.01; *** *p* ≤ 0.001.

**Figure 3 pathogens-11-00361-f003:**
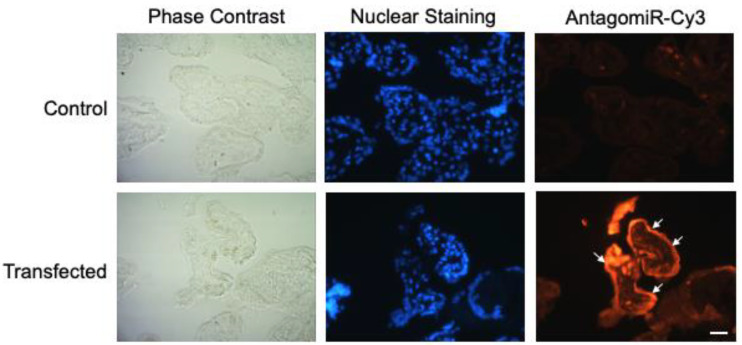
HPE transfects are localized in the trophoblast. HPE was transfected with 100 nM of AntagomiR-Cy3 for 24 h. Samples were processed using routine fluorescence methods; nuclei were stained with DAPI. White arrows indicate the fluorescence label in the trophoblast. Scale bar: 20 μm.

**Figure 4 pathogens-11-00361-f004:**
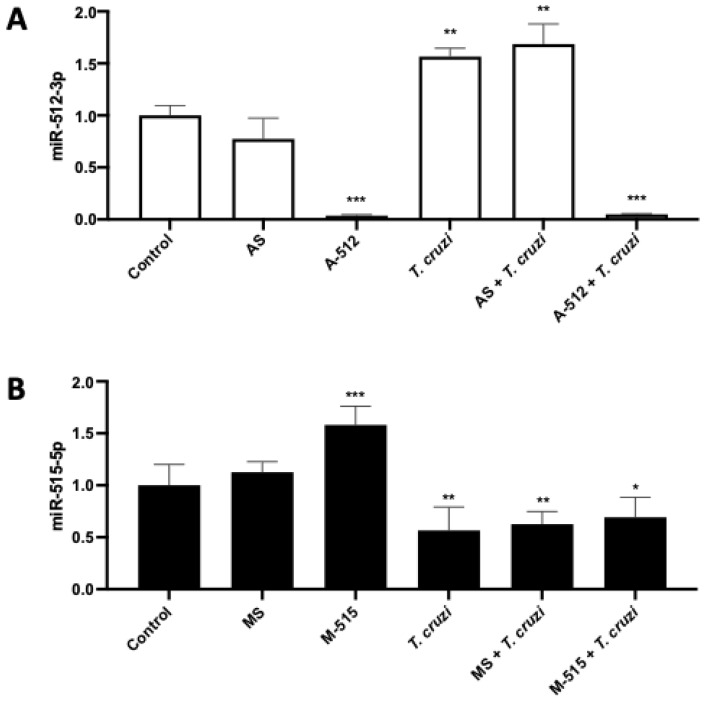
*T. cruzi* increases and decreases miR-512-3p and miR-515-5p levels, respectively, in HPE. HPE was transfected with 100 nM of miR-512-3p (**A**) or miR-515-5p (**B**) antagomirs (A-512 or A-515), mimics (M-512 or M-515), and their respective scrambles (AS or MS) for 24 h and then challenged with 10^5^
*T. cruzi* trypomastigotes for 2 h. The miRNA expression was determined by means of real-time PCR. All values are given as the means ± SD; data were normalized to control values and analyzed via one-way ANOVA and Dunnett’s post-test. * *p* ≤ 0.05; ** *p* ≤ 0.01; *** *p* ≤ 0.001.

**Figure 5 pathogens-11-00361-f005:**
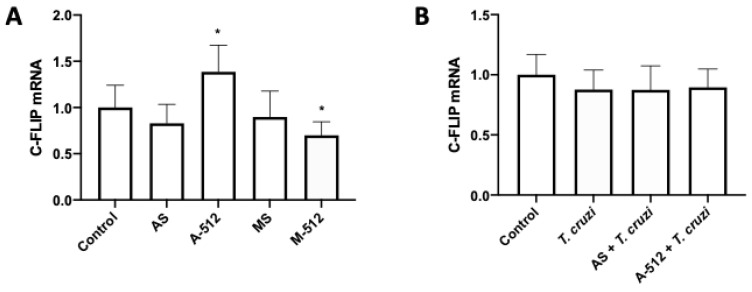
*T. cruzi* does not affect c-FLIP expression. HPE was transfected with 100 nM of miR-512-3p (**A**,**B**) antagomirs (A-512), mimics (M-512), and their respective scrambles (AS or MS) for 24 h and then challenged with 10^5^
*T. cruzi* trypomastigotes for 2 h. The c-FLIP mRNA expression was determined by means of real-time PCR. All values are given as the means ± SD; data were normalized to control values and analyzed via one-way ANOVA and Dunnett’s post-test. * *p* ≤ 0.05.

**Figure 6 pathogens-11-00361-f006:**
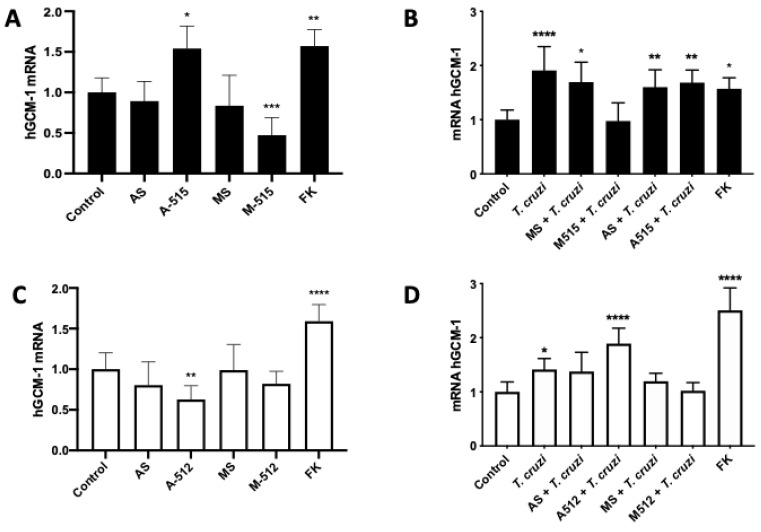
MiR-515-5p and miR-512-3p regulate *T. cruzi*-induced increase in hGCM-1. HPE was transfected with 100 nM of miR-515-5p (**A**,**B**) or miR-512-3p (**C**,**D**) antagomirs (A-512 or A-515), mimics (M-512 or M-515), and their respective scrambles (AS or MS) for 24 h and then challenged with 10^5^
*T. cruzi* trypomastigotes for 2 h. The hGCM-1 mRNA expression was determined by means of real-time PCR. All values are given as the means ± SD; data were normalized to control values and analyzed via one-way ANOVA and Dunnett’s post-test. * *p* ≤ 0.05; ** *p* ≤ 0.01; *** *p* ≤ 0.001; **** *p* ≤ 0.0001.

**Figure 7 pathogens-11-00361-f007:**
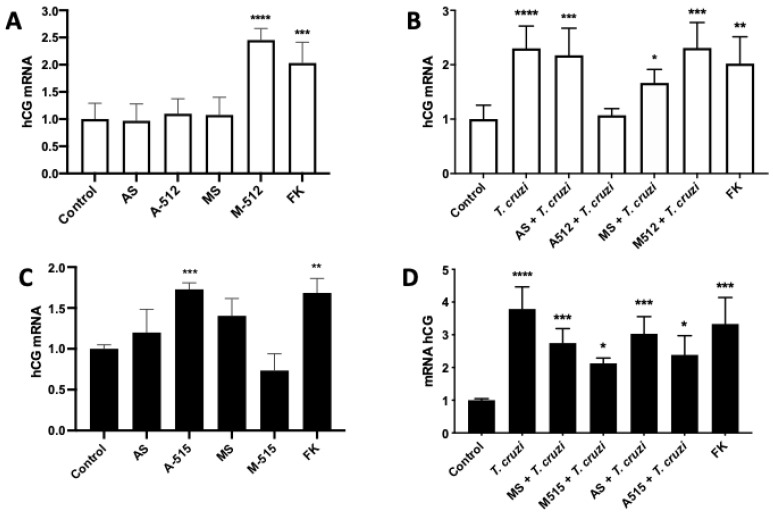
MiR-512-3p and miR-515-5p regulate *T. cruzi*-induced increase in hCG. HPE were transfected with 100 nM of miR-512-3p (**A**,**B**) or miR-515-5p (**C**,**D**) antagomirs (A-512 or A-515), mimics (M-512 or M-515), and their respective scrambles (AS or MS) for 24 h and then challenged with 10^5^
*T. cruzi* trypomastigotes for 2 h. The hCG mRNA expression was determined by means of real-time PCR. All values are given as the means ± SD; data were normalized to control values and analyzed via one-way ANOVA and Dunnett’s post-test. * *p* ≤ 0.05; ** *p* ≤ 0.01; *** *p* ≤ 0.001; **** *p* ≤ 0.0001.

**Figure 8 pathogens-11-00361-f008:**
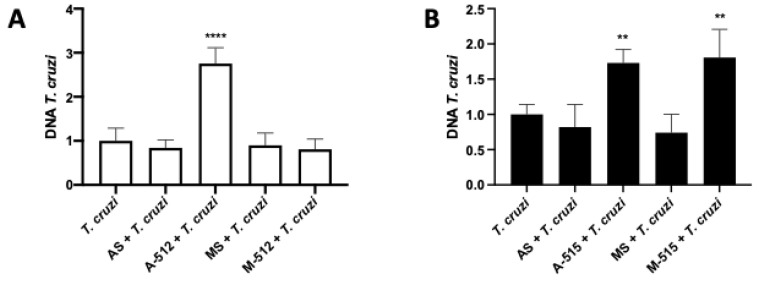
MiR-512-3p and miR-515-5p levels determine HPE susceptibility to *T. cruzi* infection. HPE was transfected with 100 nM of miR-512-3p (**A**) or miR-515-5p (**B**) antagomirs (A-512 or A-515), mimics (M-512 or M-515), and their respective scrambles (AS or MS) for 24 h and then challenged with 10^5^
*T. cruzi* trypomastigotes for 2 h. Parasite DNA was detected using real-time qPCR. Data analysis was performed using the ΔΔCt method. All values are presented as the mean ± SD (of normalized values) and analyzed via one-way ANOVA and Dunnett’st’s post-test. ** *p* ≤ 0.01; **** *p* ≤ 0.0001.

**Table 1 pathogens-11-00361-t001:** Oligonucleotides used as primers for miRNA-specific qRT-PCR analysis.

qPCR Primers	Sequence
miR-512-3p	AAGUGCUGUCAUAGCUGAGGUC
miR-515-5p	UUCUCCAAAAGAAAGCACUUUCUG
RNU6-1	GUGCUCGCUUCGGCAGCACAUAUACUAAAAUUGGAA CGAUACAGAGAAGAUUAGCAUGGCCCCUGCGCAAGG AUGACACGCAAAUUCGUGAAGCGUUCCAUAUUUU

**Table 2 pathogens-11-00361-t002:** Oligonucleotides used as primers for mRNA-specific qRT-PCR analysis.

qPCR Primers	Primer Forward	Primer Reverse	Product Length
c-FLIP	CAGGAACCCTCACCTTGTT	CGGCCCATGTAATCCTTCAT	114
hGCM1	AAGCCCTAGAAAACAATCTC	AGGTTCCATGATAAGGTCAG	148
hCG	CCCCTTGACCTGTGATGACC	TATTGTGGGAGGATCGGGGT	120
hGAPDH	AACAGCGACACCCACTCCTC	GGAGGGGAGATTCAGTGTGGT	258

**Table 3 pathogens-11-00361-t003:** Oligonucleotides used as primers for genome amplification in qPCR analysis.

qPCR Primers	Primer Forward	Primer Reverse	Product Length
*T. cruzi*	GCTCTTGCCCACAMGGGTGC	CAAGCAGCGGATAGTTCAGG	181
hGAPDH	TGATGCGTGTACAAGCGTTTT	ACATGGTATTCACCACCCCACTAT	97

## Data Availability

Data is contained within the article or Appendix A.

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
