# Peer review of "Ex Vivo Infection of Human Placental Explants by Trypanosoma cruzi Reveals a microRNA Profile Similar to That Seen in Trophoblast Differentiation"

_pathogens, 2022, doi:10.3390/pathogens11030361_

Round 1
Reviewer 1 Report
pathogens-1529921
This is an interesting contribution, considering interactions of the protozoan parasite Trypanosoma cruzi and the human placenta.
As a general comment: The authors should not abbreviate words if the abbreviation is used <10 times (e.g., CD, CCD, ST, CT, VS, NFkB, MAPK, LDH).
My suggestions are arranged according to the respective line.
21, 25: The authors should avoid an abbreviation in the Abstract.
47: The authors should delete the first part of the sentence up to … the placenta.
75: The authors should delete “show that” and begin the sentence with “In”.
101: The authors should replace “61.572” by “61.57”.
148: The authors should replace “T. cruzi trypomastigotes” by “cell culture derived trypomastigotes of T. cruzi”.
150, …: Throughout, the authors should separate the probabilities by a semicolon to avoid two closing brackets.
202: The authors should replace “-3p” by “-3p and”.
205, 206: The authors should replace “8” by “9”.
280: Throughout, the authors should include ® as superscript.
286: The authors should replace “Ypsilon” by “Y”.
305: The authors should replace “inactivated” by “complement- inactivated”.
359: The authors should replace “H20” by “H2O”.
The references contain too many mistakes. The authors should write titles in lower case and genus and species names in italics and years in bold and cruzi and gondii in lower case, correct line 405, complete lines 406 and 449, etc.
Author Response
Thank you very much for your comments, our answers are included in the attached file "Reviewer 1"

Reviewer 2 Report
This is a relatively short manuscript, which is very observational in form describing the likely influence of protozoan parasite Trypanosoma cruzi on expression of two miRNA species in human placental extracts. It's also a rather difficult manuscript to read and not obvious what the broader biological relevance is - the influence of parasite and miRNA expression on transcript levels of markers associated with trophoblast differentiation but I see no evidence of change at the protein level and no demonstration of differentiation (or cell death) per se in the explant tissue.
The study in its current form is very much as a show of data - is it submitted for a special issue of Pathogens? Whilst not a manuscript of likely high impact there may be a relevance of the data to some readers in a field which is in its infancy (i.e. the role of miRNAs at the host-parasite interface). There are, however, essential revisions indicated and two recommendations for the authors to consider.
Essential revisions
The title should accurately reflect the content of the manuscript. This in my view is not a study of trophoblast differentiation.
The abstract should be better written - it lacks the grounding context to explain why the study presented is of merit or wider interest - i.e. contextualise the incidence/prevalence of congenital Chagas disease, the relevance of the miRNAs studied (as known) and state explicitly the changes quantitated through the analyses undertaken. This manuscript in its current form is a show of data, the value for which is uncertain.
Place the y-axis for Fig 2A and Fig 2B on the same scale - as presented the data are not as transparent as they might be. It would be helpful to see expression levels in absolute amounts, rather than normalised.
Fig 3 - the phase contrast as shown is rather indistinct
Fig 4 - the data shown here are rather meaningless without the control to show the extent of LDH release as tissue (or more precisely explant) is damaged
Where change is significant, the authors should be showing in the figure which condition the change is significant to - as such the presentation is incomplete
In Section 2.6, I assume the authors are referring to Fig 9, not fig 8.
Revisions for the authors to consider
One difficulty for the reader is looking at numerous black-only bar charts with small labels - some consistent use of colour or a summary model (i.e. an additional figure) of the data shown might help the reader
Draw out better the biological relevance or wider interest of the study
Author Response
Thank you very much for your comments, our answers are included in attached file "reviewer2"

Reviewer 3 Report
In the present manuscript, the authors tried to explore the role of placenta-specific miR- 512-3p and miR-515-5p in trophoblast epithelial turnover, more specifically the trophoblast differentiation during T.cruzi infection and how overexpression for both the miRNAs mediate the placental susceptibility to ex vivo parasite infection. However, they measured the expression pattern of miR-512-3p and miR-515-5p validated target genes, i.e. hGCM-1 and c-FLIP, as well as the expression of the main trophoblast differentiation marker hCG during ex vivo T. cruzi infection of HPE cells. They also determined and how inhibition of both miRNAs affects parasite infection. The study is interesting and needs some minor corrections prior to acceptance for publication.
- The authors need to provide proper clear scale bars for figure 3 and figure 4B.
- The authors need to mention the product length of each primer they did use in qPCR experiment.
- The authors need to explain a bit more in the manuscript how they did check the HPE transfection efficiency with miRNAs. What medium they did use for transfection? They need to mention such issues in detail in the methodology section.
- The authors need to mention about the types of tissue they enrolled for histology analysis with a comprehensive writeup for the histological techniques they did adopt in the present study. How do they interpret data obtained from histological staining (H&E)? they must address these issues properly in the methodology section of the manuscript.
- Authors need to provide a few more relevant references to discuss the role of placenta-specific miR- 512-3p and miR-515-5p in trophoblast epithelial turnover and trophoblast differentiation in congenital Chagas disease.
- How did the authors confirm that enrollment of three placenta sample size will provide significant power for their present study? Why experimental data were normalized to control values? Authors need to mention the name of the test they did use for the normalization of their data and the number of control samples they did use in the present study.
- The ethics approval (approval number, year, etc.) of the study, across the study groups must be mentioned.
- Authors need to mention specifically what ANOVA test they have performed. Because Dunnett’s post hoc test is not done in all types of ANOVA.
Author Response
Thank you very much for your comments, our answers are included in the attached file "reviewer 3"

Reviewer 4 Report
Reviewer’s Comments Pathogen15299
Authors in this manuscript validated their previous findings about up and down regulation of miR-515-5P and miR-512-3P, respectively, during ex vivo infection of human placental explant (HPE) with T. cruzi. They also attempted to understand the role of these micro RNAs in T. cruzi-induced modulation of differentiation of HPE cells by assessing the levels of hCG-1 mRNA levels. However, transfection of HPE with specific mimics and antagomir of these microRNAs during T. cruzi infection produced many contradictory results. As indicated by the authors these contradictory results are possibly because HPE differentiation is mediated by a group of multiple miRNAs targeting same or different mRNAs. Overall, their results indicated that T. cruzi alters the levels of miR512-3P and miR-515-5P that somehow modulates HPE differentiation in favor of T. cruzi infection. However, it is necessary to perform further experiment to understand the exact mechanism of this processes. This reviewer thinks that the title of this article is an overstatement, which needs to be modified. Other specific comments are given below.
- Fig. 7B, the scrambled mimic with T. cruzi further increased the levels of hGCM-1 mRNAs, Why? Is it an effect of transfection?
- Antagomir-515-5P increased and mimic of 515-5P decreased hGCM-1, since hGCM-1 is the target of this miR. cruzi infection increased the levels of miR515-5P and HGCM1 both, how these could happen? Authors need to demonstrate the effect of antagomir-515-5P with T. cruzi infection on hGCM-1 expression in Fig. 7. Similarly, the effect of antagomir-515-5P with T. cruzi infection on hCG-mRNA expression also needs to be tested, which is missing in Fig. 8. These experiments could tell if the upregulation of miR-515-5P by T. cruzi is necessary for hCG upregulation and HPE differentiation.
- A similar question remains for miR-512-3P. This miRNA showed milder and indirect effect on hGCM1 expression. cruzi infection reduces the level of miR-512-3P. Author tested the effect antagomir of 512-3P with T. cruzi infection on hGCM-1 and hCG-mRNA expression (Fig. 7 and 8) but didn’t check the effect of the reverse by transfecting with the 512-3P mimic.
- Fig. 9B. Both the antagomir and mimic of miR-515-5P increased T. cruzi DNA (infection) in HPE. Authors need to provide possible explanation for these results.
Author Response
Thank you very much for your comments, we performed de requested additional experiments and oy¡ur answer are in the attached file "Reviewer4"

Round 2
Reviewer 2 Report
Manuscript improved through revision although the title remains a little overstated. There are no data to show that ex vivo infection occurs by modulating molecules involved in trophoblast differentiation - the readouts shown in the results document changes in RNA expression but phenotypic change typically correlates with changes in the abundance or activities of specific proteins and there is no data pertaining to phenotypic change in this study. A more appropriate title would be to make use of a line in the abstract:
Ex vivo infection of human placental explants by Trypanosoma cruzi reveals a microRNA profile similar to that seen in trophoblast differentiation.
Author Response
Thank you very much for your comments, we changed the title to the suggested one, please see the new version of the manuscript